# Study of the Vaginal Microbiota in Healthy Women of Reproductive Age

**DOI:** 10.3390/microorganisms9051069

**Published:** 2021-05-15

**Authors:** Melanie C. Alonzo Martínez, Eduardo Cazorla, Esther Cánovas, Juan F. Martínez-Blanch, Empar Chenoll, Eric Climent, Vicente Navarro-López

**Affiliations:** 1PhD Program in Health Sciences, Catholic University of Murcia UCAM, Campus de los Jerónimos n°135, 30107 Murcia, Spain; melaniealonzo@gmail.com; 2University Hospital of Torrevieja, Carretera CV 95, s/n, 03186 Alicante, Spain; ecanovas@torrevieja-salud.com; 3MiBioPath Group, Catholic University of Murcia, Campus de los Jerónimos n°135, 30107 Murcia, Spain; vnavarro@vinaloposalud.com; 4Archer Daniels Midland Co-Biopolis, 46980 Valencia, Spain; juan.martinezblanch@adm.com (J.F.M.-B.); maria.chenoll@adm.com (E.C.); eric.climent@adm.com (E.C.); 5University Hospital of Vinalopó, Carrer Tonico Sansano Mora, 14, 03293 Alicante, Spain

**Keywords:** vaginal microbiota, dysbiosis, *Lactobacillus*, *Lactobacillus crispatus*, microbial communities, 16S Ribosomal RNA

## Abstract

Understanding the characteristics of the vaginal microbiota of our patients allows us to carry out both a personalized therapeutic approach and a closer follow-up in those with microbiota susceptible to dysbiosis. This trial pursues the analysis of the vaginal microbiota of premenopausal women and its fluctuations within a four-week follow-up period. Vaginal samples of 76 fertile women were taken at a baseline visit and at a final visit (day 28 ± 5). To perform a phylogenetic study, we employed massive sequencing techniques to detect the 16S rRNA gene of the vaginal microbiota. The most prevalent vaginal microbial community was type I (34.87%), dominated by *Lactobacillus crispatus*. Vaginal microbial community types II (*Lactobacillus gasseri*) and V (*Lactobacillus jensenii*) were underrepresented in our population. When repeating the sampling process four weeks later, 75% of our patients maintained their initial bacterial community. In the follicular phase, the most recurrent microbiota was type III (*Lactobacillus iners*); in the periovulatory phase, types III and IV (microbial diversity); finally, in the luteal phase, the most frequent type was IV. The most prevalent vaginal bacterial community in our population was dominated by *L. crispatus*. The vaginal microbiota was resistant to changes in its bacterial community in 75% of our patients, even between consecutive menstrual cycles.

## 1. Introduction

Vaginal microbiota can be classified into five types based on its composition [1]. Types I, II, III, and V are dominated by *Lactobacillus crispatus*, *L. gasseri*, *L. iners*, and *L. jensenii*, respectively. Type IV microbiota include a wider variety of microorganisms, with fewer *Lactobacillus* species and a prevalence of anaerobic bacteria [1].

*Lactobacilli* living in the vagina protect their environment from pathogens in diverse ways. First, they produce antimicrobial compounds such as bacteriocins, which inhibit bacterial multiplication and promote their destruction [2]. Second, they release biosurfactants, which inhibit the binding of pathogenic bacteria and hinder the formation of biofilm [3]. Third, they congregate with pathogenic microorganisms to both prevent the latter from adhering to the vaginal epithelium and to remove them with microbicidal proteins [4]. Fourth, *lactobacilli* protect the integrity and functionality of the vaginal epithelium and its mucin layer, making the environment less susceptible to pathogens [5]. Fifth, the adhesion of this genus to the vaginal epithelium plays a key role in vaginal colonization and inhibiting microbial adherence [4,6]. Sixth, they produce lactic acid by glycogen fermentation of the vaginal epithelium, which is exposed to estrogens, generating an acidic pH and increasing the viscosity of the vaginal discharge. Seventh, *Lactobacillus* exert cytotoxic activity against cervical cancer cells as evidenced in two in vitro studies [7,8]. Eighth, some *lactobacilli* release hydrogen peroxide, which serves a protective role against pathogenic microorganisms [2,9]. Lastly, *lactobacilli* compete against pathogenic microorganisms for the nutrients available in the vaginal ecosystem [10].

The vaginal microbiota of premenopausal healthy women is normally governed by the *Lactobacillus* genus. The predominant species are *L. crispatus* and *L. iners*, with fewer numbers of other microorganisms [11,12]. Many of these species have been discovered via new 16s Ribosomal RNA gene sequencing techniques (16S rRNA), with the 16S rRNA being a component of prokaryotic cells such as bacteria. These techniques are used to phylogenetically analyze microbiota, detect bacterial taxa that cannot be cultured in standard media, and classify and quantify thousands of microorganisms at the same time.

In recent years, new evidence has emerged suggesting that the status of the vaginal microbiota plays a particularly important role in the persistence and elimination of pathogenic microorganisms. Currently available treatments for vaginal dysbiosis, such as the use of antimicrobial agents, encounter high and increasing percentages of resistance, resulting in chronic inflammation and recurrent infections, which may affect the quality of life of patients. The aim of this study is to describe the vaginal microbiota of premenopausal women, to analyze the most predominant bacterial communities in this population, and to examine their individual and collective variations over a 30-day follow-up period.

## 2. Materials and Methods

A prospective observational study of the vaginal microbiota and its inter- and intraindividual variability was performed over a four-week period, conducted in a cohort of healthy women of reproductive age. The first 80 patients visiting the General Gynecology and Lower Genital Tract Unit at the University Hospital of Torrevieja who fulfilled all the inclusion criteria and none of the exclusion criteria were selected. Inclusion criteria involved women of reproductive age (≥30 years) with no clinical diagnosis of bacterial vaginosis, and who agreed to participate in the study. Exclusion criteria included women who were pregnant, breastfeeding, using contraceptives, reluctant to use male contraceptive methods, using spermicides, and/or had taken probiotics within the last two months or antibiotics within the last two weeks. At the baseline visit, all patients received the patient information sheet, in which the methodology of this voluntary study was explained; the patients signed an informed consent. Thereafter, they underwent a urine pregnancy test and a speculum examination to rule out any signs of vaginosis. In the absence of any pathological findings, the first vaginal sample was collected with a swab by moving it on the vaginal walls for 20 s. The last visit took place after 28 ± 5 days, which corresponded to the next menstrual cycle. At this visit, a second vaginal sample was collected using the methodology described above.

The samples were immediately preserved in a collection tube containing 1 mL of bacterial DNA and RNA stabilizing solution (OMNIgene VAGINAL, DNA Genotek, Otawa, ON, Canada) at −30 °C for a maximum of 6 months before analysis. The samples were taken to the laboratory in a temperature-controlled container.

Highly pure DNA was obtained using a commercial kit (DNeasy Blood & Tissue Kit, Qiagen, Hilden, Germany). Genomic libraries were built using PCR products obtained from the hypervariable V3–V4 region of the 16S rRNA gene [13], following the Illumina *16S rDNA gene Metagenomic Sequencing Library Preparation protocol* (Illumina, San Diego, CA, USA). Sequencing was performed with Illumina *MiSeq Sequencer* 300PE to obtain at least 50,000 reads per sample.

The obtained sequences were assigned to each sample according to their barcoding. “Forward” and “reverse” sequences were merged using BBmerge software to obtain a representative sequence for each pair [14]. The average length of the merged sequences was 450 nucleotides. Then, 16S rRNA gene amplification adapters were eliminated with cutadapt v2.6 software [15]. A quality control was then performed using BBtools’ reformat software, and sequences with a mean quality lower than 20 in the Phred scale and with a length bellow 200 nucleotides were eliminated. FASTQ files were converted into FASTA files, and the 4.8 version of the cd.hit-dup software was used to delete any chimera that may appear during the amplification and sequencing step [16]. BLAST analysis was then performed on the NCBI-16S-rRNA gene database using blastn version 2.10.0+. The resulting XML files were processed using a python script, developed internally to record each sequence at different phylogenetic levels (phylum, family, genus, and species). Alpha and beta diversity analyses were performed using the “vegan” library of the R 3.6 version and Bray–Curtis distances. The differential presence of taxa was analyzed using the R DESeq2 library [17].

Subsequently, the results obtained at different taxonomic levels (phylum, family, and genus) were compared to detect their specific bacterial populations. Bacterial diversity was estimated using alpha diversity tests (species richness, Shannon and Simpson diversity index) and beta diversity tests (Local Contribution to Beta Diversity, PERMANOVA test over the Principal Coordinates Analysis variables or PCoA, and the beta dispersion analysis by variable pairs).

In this study, we paid special attention to the evolution of the *Lactobacillus* genus as an indicator of a healthy vaginal microbiome. To analyze pathogenic groups and to compare our findings with previously reported results [1], we also performed a species estimate, given the insufficient amplification size sequenced by this technique. The pathogenic bacterial groups included *Gardnerella vaginalis*, *Atopobium vaginae*, *Streptococcus agalactiae*, and *Cutibacterium acnes* (formerly: *Propionibacterium acnes*). Moreover, samples were classified into bacterial communities according to Ravel et al. [1], and the changes between both visits were assessed. This study was conducted according to the Declaration of Helsinki and the protocol was approved by the Ethics Committee for Research of University Hospital of Torrevieja and University Hospital of Vinalopó (registration No. 2018/01). Data were recorded in a Data Collection Notebook with patient coding pursuant to the European Union Data Protection Regulation 2016/679.

## 3. Results

### 3.1. Sample Collection

To reach the target sample size, 201 women were assessed, of whom 121 were excluded because they did not fulfill the inclusion criteria. Recruitment was performed from November 2018 to May 2019 and ended when an 80-patient sample was obtained. The patients in our study were between 30 and 49 years old, with a mean age of 40 years. They were healthy with no notable comorbidities. During the follow-up period, four patients withdrew from the study; two because they did not attend the final visit and two due to errors during sample processing. Therefore, there were 76 patients with valid vaginal samples taken at the beginning and at the end of the study.

### 3.2. Bioinformatic Analysis

Following the bioinformatic analysis of samples, library amplification performance was optimal. In terms of genus, rarefaction curves based on the sequences obtained showed that curve saturation was reached, identifying practically all genera in the samples (Figure 1). On increasing the number of sequences (horizontal axis) analyzed, more genera were detected (vertical axis, Operational Taxonomic Unit (OTU)), until reaching a point at which the number of genera detected stopped increasing, regardless of the number of sequences analyzed, thus arriving at a plateau.

On comparing the findings of both visits, no differences were observed for the *Lactobacillus* genus (non-parametric Wilcoxon test, W = 746, *p* = 0.7908). No significant variations between both visits were observed for the selected pathogenic species. Results of the Wilcoxon test conducted on these species yielded were as follows: *G. vaginalis* (W = 665, *p* = 0.5728), *A. vaginae* (W = 730.5, *p* = 0.9171), and *S. agalactiae* (W = 706.5, *p* = 0.8862). *Cutibacterium acnes* was not detected in any of the samples analyzed. Unification of these microorganisms into the “pathogens” group did not yield significant values (W = 628, *p* = 0.3432).

#### 3.2.1. Alpha and Beta Diversity Tests

Alpha diversity (sample richness and Shannon and Simpson diversity index) and beta diversity tests conducted at the phylum, family, and genus levels did not show statistically significant differences (Figure 2). Sample diversity was relatively homogeneous, and no sample stood out at any taxonomic level. Furthermore, after maximizing the differences observed among groups, a Canonical Correlation Analysis showed that no combination of existing taxonomic levels was able to determine differences between the visits.

#### 3.2.2. Results by Taxonomic Group

Sample analysis using DESeq2 showed that the most important phylum was Fusobacteria, when comparing the baseline visit with the final visit. This was the only phylum whose values significantly decreased at the final visit (adjusted *p*-value = 0.001). The most frequently observed phylum in our population was *Firmicutes*, followed by *Actinobacteria*, *Bacteroidetes*, *Proteobacteria*, and *Tenericutes*.

A decrease was observed in the *Fusobacteriaceae* family (adjusted *p*-value = 0.0006), whereas an increase was found in the *Enterobacteriaceae* family (adjusted *p*-value = 3.7220 × 10^−6^); both changes were statistically significant at the final visit. In general, the most frequently isolated family was *Lactobacillaceae*, followed by *Bifidobacteriaceae*, *Prevotellaceae*, and *Atopobiaceae*.

At genus level, a statistically significant depletion in the *Parvimonas* (adjusted *p*-value = 1.7935 × 10^−7^), *Megasphaera* (adjusted *p*-value = 4.2098 × 10^−6^), *Fusobacterium* (adjusted *p*-value = 0.0003), *Mageeibacillus* (adjusted *p*-value = 0.0355), and *Mycoplasma* (adjusted *p*-value = 0.0421) genera was observed. However, there was a statistically significant increase in *Staphylococcus* (adjusted *p*-value = 0.0421), *Actinomyces* (adjusted *p*-value = 0.0011), and *Escherichia* (adjusted *p*-value = 1.5501 × 10^−17^) at the final visit (Figure 3 and Figure 4). The most frequently observed genus was *Lactobacillus*, followed by *Gardnerella*, *Prevotella*, and *Atopobium*.

Finally, a species-level approximation was performed, finding a statistically significant increase for *Lactobacillus acidophilus* (adjusted *p*-value = 0.0063), *L. psittaci* (adjusted *p*-value = 0.0011), *Actinomyces neuii* (adjusted *p*-value = 5.0435 × 10^−6^), *Prevotella bivia* (adjusted *p*-value = 1.6449 × 10^−7^), and *Escherichia fergusonii* (adjusted *p*-value = 8.2095 × 10^−17^) species. Furthermore, a significant depletion was observed for *Parvimonas micra* (adjusted *p*-value = 2.4315 × 10^−7^), *Peptoniphilus lacrimalis* (adjusted *p*-value = 0.0038), *Peptoniphilus coxii* (adjusted *p*-value = 0.0099), and *Mageeibacillus indolicus* (adjusted *p*-value = 0.0314) species. The most abundant species were *Lactobacillus crispatus* and *L. iners*.

#### 3.2.3. Vaginal Microbial Communities

The bacterial communities most frequently observed at both visits were type I and type III, with 53 (34.87%) and 50 (32.89%) samples obtained, respectively. They were followed by type IV microbiota, with 39 (25.66%) samples obtained. Type II and V communities were found in eight (5.26%) and two (1.32%) samples, respectively (Figure A1).

Table 1 shows the transition between the bacterial communities following analysis of the database obtained from the baseline visit until the final visit, as well as, for each microbiota type, the percentage of the total found at the baseline visit. About 75% of patients maintained their baseline microbiota type during the follow-up, whereas only 25% showed changes in their bacterial community composition. Microbiota types I, III, and IV were maintained in up to 78% of cases between the visits. The number of patients with types II and V was scarce.

Finally, bacterial communities were determined based on the menstrual cycle phase in which the baseline sample was collected. In the follicular phase, the most prevalent microbiota was type III (57.14% of samples), followed by type IV (38.89%). In the periovulatory phase, type III and IV communities were mostly observed (14.29% and 11.11%, respectively). Finally, type IV (44.44%) and I (36%) microbiota were those most frequently found in the luteal phase.

## 4. Discussion

The vaginal ecosystem is complex, and its composition may dramatically change depending on various factors that may cause an imbalance. Internal factors include menstruation, postpartum, menopause, diabetes, genetic polymorphisms, age, and race [1,18]. External factors include smoking, vaginal douching and other hygiene habits, recent sexual intercourse, number of sexual partners and practices, use of spermicides, and use of medications such as oral contraceptives, antibiotics, and immunosuppressants [6,18].

To ensure that the vaginal microbiota was not affected by external factors, we only included patients who had not taken any contraceptives or antibiotics in the previous two weeks, and/or probiotics within the previous two months. Following the analysis of the data collected, the most recurrent vaginal bacterial communities at both visits were type I and type III, which mainly comprised *Lactobacillus crispatus* and *L. iners*, respectively. These were followed by type IV microbiota, which are characterized by bacterial diversity without *Lactobacillus* predominance. Although Ravel et al. [1] reported a prevalence of type IV community in asymptomatic Hispanic women, their study was performed in a cohort of American patients. This type of microbiota can be considered as a physiological variable in this ethnic group, as well as in Africans, although type IV microbiota are also associated with a higher incidence of vaginosis and a four-fold higher risk of contracting *human immunodeficiency virus* infection [1,19]. However, the results of our study are comparable to those reported by Ravel et al. [1] and Gajer et al. [20] for American patients. These data further highlight the importance of studying the vaginal microbiota of each population.

Similar to the observations by Gajer et al. [20], 75% of patients maintained their baseline bacterial community during the follow-up period. Regarding types II (*L. gasseri*) and V (*L. jensenii*) microbiota, as the number of patients in these groups was very low, results cannot be generalized. In parallel, Romero et al. [21] reported an underrepresentation of types II and V microbiota in American women. Generally, we observed a tendency to resist changes in the baseline microbiota of our patients. The microbiota tend to remain stable over time, with low intraindividual variability, although it may undergo transient alterations triggered by external and hormonal factors, and subsequently return to basal levels. In this respect, our sample follows the *Community resilience* hypothesis [1]. Moreover, as the samples were collected during different menstrual cycles, we can conclude that although menstruation may alter the microbiota, these alterations were not sufficient to change the bacterial community of most of this study’s patients. Our findings are consistent with those reported by Kyono et al. [22], who found a short-term stable *Lactobacillus* concentration both within the same cycle, and between different cycles, in a small cohort of healthy patients.

Finally, we compared the menstrual cycle phase in which samples were taken at the baseline visit with the bacterial community found. We observed that type III microbiota were most prevalent in the follicular phase samples, whereas type III and IV communities presented similar numbers in the periovulatory phase, while type IV microbiota were mostly present in the luteal phase. There are studies supporting the idea that peaks in estrogen during the menstrual cycle (periovulatory and luteal phases) enhance vaginal microbiota stability, and promote *Lactobacillus* predominance, by regulating glycogen availability in the vaginal epithelium [20]. However, the resulting microbiota is the product of a combination of several modulating and modifying internal and external factors.

The way in which samples are analyzed is a key factor for the study of vaginal microbiota. Many published clinical trials have studied vaginal microbiota using culture-based methods that exclude a significant number of species, such as *L. iners*, which do not grow in most *Lactobacillus* culture media [2,23]. Moreover, some studies classify vaginal microbiota’s status based on either the Amsel criteria or the Nugent score, which may both have some interobserver variability. In our study, samples were analyzed using massive sequencing techniques aimed at detecting the 16S rRNA gene to objectively identify all bacteria, thus eliminating any bias caused by growth capacity in a medium and analyst subjectivity. However, given the amplification size sequenced by these techniques, identifications at the species level are considered approximations.

This is one of the few published studies to include an accurate description of the vaginal ecosystem composition of a significant number of samples taken from premenopausal patients, and the changes observed within a four-week period. By understanding and analyzing the characteristics of the vaginal microbiota of our patients, we can implement a personalized therapeutic approach in our clinical practice and closely monitor patients whose microbiota is susceptible to dysbiosis. Thus, further research is warranted to gain greater insight into the effect of exogenous administration of the most predominant vaginal bacteria for the treatment and prevention of vaginal dysbiosis.

## Figures and Tables

**Figure 1 microorganisms-09-01069-f001:**
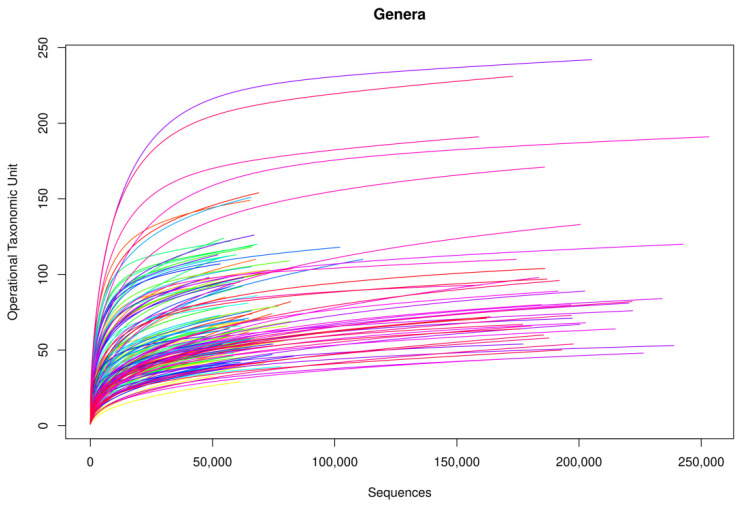
Rarefaction curves of the amplified samples for bacterial detection at a genus level.

**Figure 2 microorganisms-09-01069-f002:**
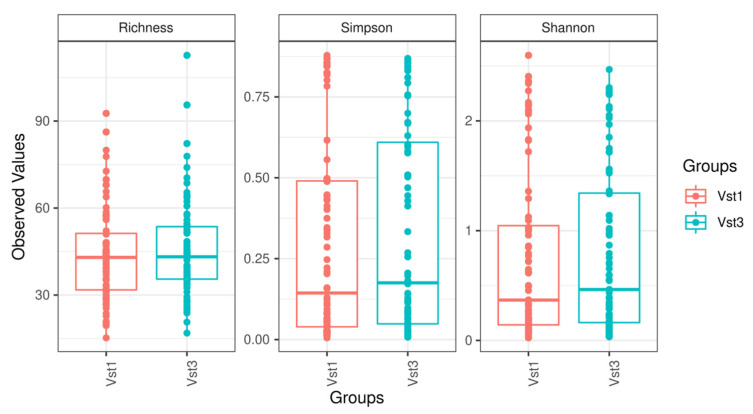
Boxplot figures representing sample diversity based on the variables studied at each visit. Vst1: baseline visit, red. Vst3: final visit, blue.

**Figure 3 microorganisms-09-01069-f003:**
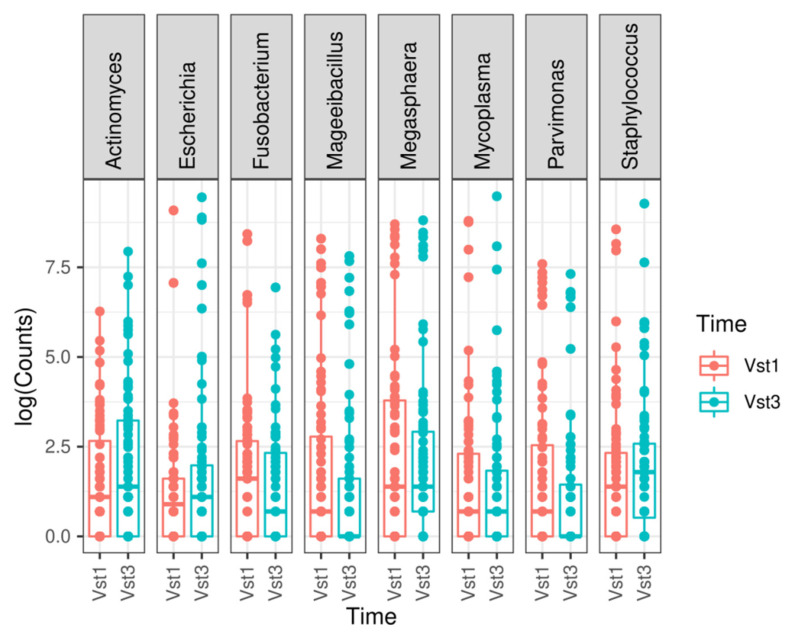
Most relevant genera found when comparing both visits. A boxplot is shown for each statistically significant genus. Vst1: baseline visit, red. Vst3: final visit, blue.

**Figure 4 microorganisms-09-01069-f004:**
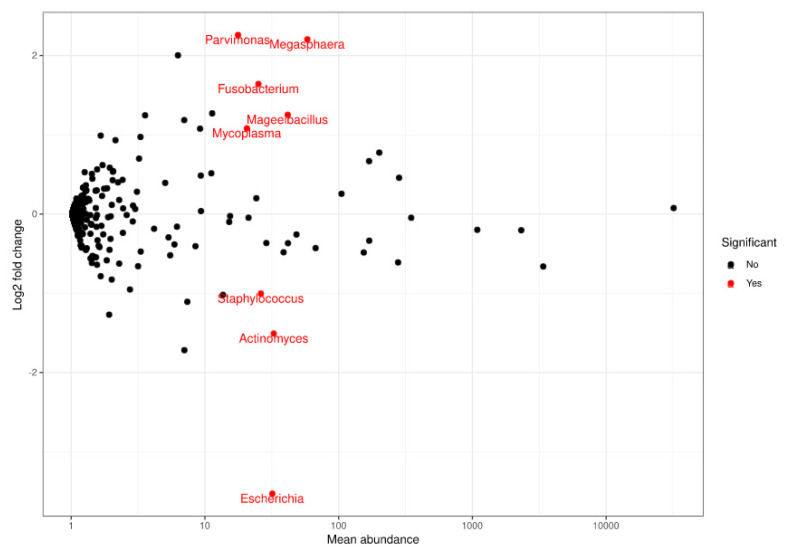
Representation of FoldChange values and computation of the sequences obtained when comparing populations at a genus level. Red dots: genera that significantly decreased (upper end) or increased (lower end) at the final visit based on the comparison performed using DESeq.

**Table 1 microorganisms-09-01069-t001:** Transition between vaginal bacterial communities, n (%).

Transition between Vaginal Bacterial Communities	I	II	III	IV	V
I	19 (76)	0	1 (4)	5 (20)	0
II	0	4 (100)	0	0	0
III	7 (25)	0	20 (71.43)	1 (3.57)	0
IV	2 (11.11)	0	1 (5.56)	14 (77.78)	1 (5.56)
V	0	0	0	1 (100)	0

First column: baseline visit. First line: final visit.

## Data Availability

The data presented in this study are available on request from the corresponding author. The data are not publicly available due to privacy restrictions.

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
