# Peer review of "Study of the Vaginal Microbiota in Healthy Women of Reproductive Age"

_microorganisms, 2021, doi:10.3390/microorganisms9051069_

Round 1

Reviewer 1 Report

The paper by et al. Martínez provides a well presented and detailed overview on the composition of the vaginal microbiota of 76 premenopausal healthy women over a 4 weeks period. VAginal samples were collected at baseline and after 28+/-5 days. Metagenomic Sequencing was performed for the phylogenetic study, bacterial diversity was evaluated by alpha and beta diversity tests. A stability of the vaginal microbiota composition in 75% of the patients was observed. The results were compared to analogue studies. Moreover, bacterial composition at baseline was related to the menstrual phase.

The study is appropriately designed, the results are reported clearly and properly examined in the discussion. The quality of english is good. The figures have good quality.

The authors should provide the demographics and the characteristics of the patients (age, BMI, previous deliveries, race/ethnicity, and, possibly, others like alcohol use, smoking habits, comorbidities, drugs). 

Few minor comments:

  • the introduction part regarding lactobacilli is too long and need to be shortened
  • in the abstract it is reported that the microbiota of 80 patients was analysed, while in the results only 76 patients completed the study; this numbers should be corrected
  • since 25% of the patients went through a compositional bacterial change, it cannot be strongly assessed that "the vaginal microbiota is resistant to a change of its bacterial community, even between consecutive menstrual cycles" as reported in the abstract because this proportion is relevant

Reviewer 2 Report

Overall, this manuscript is good. However, it needs an extensively english correction. I do suggest a revision by a native english editing company. There are many mistakes such as punctuantion, grammar and written.

The introduction is very informative but becomes hard to read based on the bad english. 

The methodology must be better described. The current frame of it is quite confusing.

all the figures are in bad quality. It is impossible to understand the caption inside the picture cause' they're too small. All the figure must be substantially improved.

The results and discussion are ok, it could better described. 

my major concer is about this following conclusion "By understanding the characteristics of the vaginal microbiota of our patients, we can achieve a personalized therapeutic approach and a closer monitoring of patients whose microbiota is susceptible to dysbiosis''. 

the personalized therapeutic approch is only useful to those patients from this study. You cannot exploit that to everyone. This a very complicated information. 

Round 2

Reviewer 2 Report

This manuscript is now suitable for publication.